# Healthy Zerumbone: From Natural Sources to Strategies to Improve Its Bioavailability and Oral Administration

**DOI:** 10.3390/plants12010005

**Published:** 2022-12-20

**Authors:** María Dolores Ibáñez, Noelia M. Sánchez-Ballester, María Amparo Blázquez

**Affiliations:** 1Departament de Farmacologia, Facultat de Farmàcia, Universitat de València, Avd. Vicent Andrés Estellés s/n, Burjassot, 46100 València, Spain; 2ICGM, Département Chimie et Matériaux Moléculaires, University of Montpellier, CNRS, ENSCM, 34090 Montpellier, France; 3Department of Pharmacy, Nîmes University Hospital, 30900 Nîmes, France

**Keywords:** zerumbone, agro-industrial applications, encapsulation, bioavailability, oral delivery

## Abstract

Zerumbone is a multifunctional compound with antimicrobial, antitumor, hyperalgesic, antioxidant and anti-inflammatory applications, and constitutes a point molecule for the future synthesis of derivatives with improved efficiency. This monocyclic sesquiterpenoid is found in high content in wild ginger (*Zingiber zerumbet* Smith), a perennial herb with economic importance as an ornamental as well as a medicinal plant. The presence of zerumbone is a distinctive feature that allows identification and differentiation from other species, not only in *Zingiber*, but also in *Curcuma*, *Alpinia*, *Boesenbergia*, *Ethlingera* and *Ammomum* spp., as well as related families (Costaceaee). To successfully use zerumbone in areas such as medicine, food and agriculture, further research on improving its low solubility and bioavailability, as well as its preservation, is a major current priority. In addition, despite its promising pharmacological activities, preclinical and clinical studies are required to demonstrate and evaluate the in vivo efficacy of zerumbone.

## 1. Introduction

Zerumbone (2,6,9,9-tetramethyl-(2E,6E,10E)-cycloundeca-2,6,10-trien-1-one) is a highly volatile monocyclic sesquiterpene compound (C_15_H_22_O) found in large amounts in the rhizome of wild edible ginger, *Zingiber zerumbet* (L.) Roscoe ex Sm. essential oil [1,2]. It is mainly distributed in tropical and subtropical regions, including Sri Lanka, Nepal, Bangladesh, Malaysia, India, and southwest China (commonly known as *hong qiu jiang*). Traditionally, this rhizomatous herbaceous species, belonging to the Zingiberaceae family, is also known as bitter ginger, shampoo ginger and pinecone ginger. It has applications as a flavouring and appetizing agent and for treating pain in folk medicine [3,4]. The genus *Zingiber* comprises about 141 species (12 species native to China), mainly cultivated in Asia, Central, South America and Africa, and is an important source of essential oil widely used in the perfumery, cosmetics, and pharmaceutical industries. Although the different members of this genus have similar morphology, they differ significantly in their pharmacological and therapeutic properties [5,6,7]. Among these species, many studies have focused on five species: *Zingiber officinale* Rosc (ginger); *Zingiber zerumbet* (L.) Smith, *Zingiber corallinum* Hance, *Zingiber mioga* (Thunb.) Rosc and *Zingiber striolatum* Diels. The main compounds obtained from the essential oils of these species are considerably different, with α-zingiberene, zerumbone and palmitic acid extracted as the main component of *Z. officinale*, *Z. zerumbet* and *Z. striolatum*, respectively. Focusing our attention on *Z. zerumbet*, the main bioactive compound found in the ethanolic extract and essential oil is zerumbone [8]. However, the chemical composition of *Z. zerumbet* rhizome essential oil is different depending on the geographical origin and the drying processes used, which can reduce the biological activity. In this sense, the zerumbone content in essential oils obtained from fresh rhizomes from different geographical origins varies from 8.1 to 84.8%, whereas in essential oils from dried rhizomes, the zerumbone content ranges from 1.2 to 35.5% [6]. Zerumbone (69.9%), followed by α-humulene (12.9%), ae also the main compounds identified in the essential oil of a variety of *Z*. *zerumbet,* identified as *Z*. *zerumbet* var. *darcyi*, which grows wild and is used against coughs and colds by the Manipur tribes in northeast India [9]. A similar composition of zerumbone (72.9%) and α-humulene (7.1%), was found in fresh rhizomes of *Z. zerumbet* collected in an experimental culture in Uttarakhand [2]. However, the amount of zerumbone (40.2%) was lower when the essential oil was obtained from the rhizome collected in Xishuangbanna (Yunnan Province, China) [10]. Zerumbone is a multifunctional compound that exhibits various biological activities, such as antimicrobial, antioxidant, anti-inflammatory, antitumor or antihyperalgesic, attributed to the structure of this bioactive compound biosynthesized from α-humulene (Figure 1) [11]. This naturally occurring sesquiterpene has a 65.3 °C melting point, contains an 11-membered ring, an α, β-unsaturated carbonyl group, and three double bonds (two conjugated and one isolated) at C6, C2 and C10 as part of a cross-conjugated system, which result in its pharmacological activity. It is interesting to note that α-humulene is a versatile starting material for conversion to other derivatives and therapeutic compounds such as paclitaxel, which is widely used in breast cancer treatment.

In the zerumbone biosynthetic path, the enzyme cytochrome P450 hydroxylates α-humulene at the C8 position, and the dehydrogenation of 8-hydroxy humulene is catalyzed by a short-chain dehydrogenase. Zerumbone has also been biosynthesized from metabolically modified *Saccharomyces cerevisiae* by introducing α-humulene synthase, α-humulene 8-hydroxylase and zerumbone synthase variant together with *Arabidopsis thaliana* cytochrome P reductase into a yeast strain [12]. The α, β-unsaturated carbonyl group of zerumbone is thought to be responsible for its cell membrane rupturing effect, explaining the mechanism of action of zerumbone against methicillin-resistant *Staphylococcus aureus* (MRSA). In fact, zerumbone exerts anti-MRSA effects by causing membrane depolarization, increasing membrane permeability, and ultimately disrupting the cell membrane and killing the bacteria [13]. Zerumbone possesses significant antimicrobial activity against *Aspergillus niger*, *Bacillus cereus*, *B. subtilis*, *Bacteroides fragilis*, *Enterococcus faecalis*, *Escherichia coli*, *Helicobacter pylori*, *Proteus vulgaris*, *Salmonella choleraesuis*, *S. typhimurium*, *S. aureus*, *S. epidermitis*, *Streptococcus mutans* and *Yersinia enterocolitica*. It has an anti-virulence effect by inhibiting the biofilm formation and hyphal growth of *Candida albicans* in a concentration-dependent manner. In addition, it exhibits significant cytotoxic activity against lung, colon, leukemia, ovarian, skin, liver and breast cancers, with selective cytotoxic activity on human tumor cell lines, not found with the structural analogue, α-humulene, that lacks the carbonyl group (Figure 1) [4,5,6,13,14]. Several studies have reported the antitumor activity of zerumbone in vitro, as well as in vivo models such as breast, colorectal, cervical, lung, renal cell carcinoma and skin cancers [4]. Furthermore, zerumbone is able to resensitize breast cancer cells to paclitaxel, increasing oxidative stress [15]. The combination of zerumbone-vinblastine and zerumbone-paclitaxel was strongly synergistic in the inhibition of HeLa cell proliferation, showing a similar mechanism of action to other clinically used chemotherapeutic drugs [16], and the combination with cisplatin showed a dose-dependent induction of cytotoxicity and apoptosis in head and neck squamous cell line carcinoma [17]. The anti-biofilm and antimicrobial effects of zerumbone have been studied against the main anaerobic opportunistic pathogen of ulcerative colitis and other inflammatory bowel diseases, *B. fragilis*. This oxygenated sesquiterpene significantly inhibited biofilm formation, disrupted established biofilms as well as microbial growth of *B. fragilis*, and is an antipathogenic compound capable of preventing chronic inflammatory diseases of this microorganism [18]. Similar effects were observed at sub-inhibitory doses of zerumbone for the multidrug-resistant opportunistic human pathogen *Acinetobacter baumannii* [19]. Zerumbone is also, in a concentration-dependent manner, an antimicrobial and antibiofilm compound for the management of nosocomial infections against mono-species and dual-species biofilms of *C. albicans* and *S. aureus* [20]. Furthermore, zerumbone is not only capable of influencing microbial richness and diversity but can also improve the balance of intestinal microbial composition. In other words, zerumbone can restore the composition of the intestinal microbiota associated with various diseases such as inflammatory bowel diseases and colon cancer [21]. The anti-inflammatory activity of zerumbone is mainly due to the inhibition of pro-inflammatory genes and antioxidant pathways. Zerumbone was demonstrated to inhibit cyclooxygenase, prostaglandin and interleukin and was able to protect the gastric mucosa via antioxidant mechanisms [22]. Regarding hyperalgesic properties, zerumbone significantly alleviated tactile and cold allodynia, as well as mechanical and thermal hyperalgesia [23]. Zerumbone was able to exert antiallodynic and anti-hyperalgesic effects in a mouse model of neuropathic pain through inhibition of the serotonergic system, specifically 5-HT receptor subtypes 1A 1B, 2A, 3, 6 and 7 [24]. More recent studies indicate that in the antiallodynic and anti-hyperalgesic effects of zerumbone, K^+^ channels and µ-, δ- and κ-opioid receptor subtypes [25], as well as cannabinoids and peroxisome proliferator-activated receptors (PPARs), are involved [26] in its modulation of neuropathic pain. In addition, the anti-hyperglycemic and anti-hyperlipidemic effects of zerumbone have beneficial effects in reducing the risk of diabetic retinopathy [27], and the dual acetylcholinesterase and butyrylcholinesterase inhibitory activity of zerumbone with high permeability across the blood-brain barrier supports its benefits as a preventive agent in Alzheimer’s disease [28].The pharmacokinetic properties of zerumbone explain the low bioavailability and distribution in the body of this water-insoluble compound. To achieve therapeutic effects in topical applications, incorporation of zerumbone into liquid or semi-solid lipid-based carriers, such as nanostructured lipid gels, improves its solubility and availability, promoting wound healing [29,30]. The physicochemical properties of zerumbone-loaded nanostructured lipid carriers were shown to be relatively stable when stored at 4 °C, 25 °C or 40 °C for at least one month, making them an interesting sustained-release drug carrier for the treatment of leukemia [31]. Nanosuspensions are also used to improve the oral bioavailability of poorly aqueous soluble drugs. In this sense, a nanosuspension formulated using high-pressure homogenization with sodium dodecyl sulphate and hydroxypropyl methylcellulose improved the saturation solubility and dissolution profile of zerumbone [32], and its inclusion complexes with several cyclodextrin derivatives, were also useful for improving the solubility of zerumbone [33]. A clinical study of the inclusion complex of zerumbone with hydroxypropyl-β-cyclodextrin revealed that 20 µM of zerumbone injected with 0.15 g of ranitidine orally for a 20-day treatment was quite effective and safe in acute and chronic gastritis, with or without *H. pylori* infection [34]. Thus, many studies have focused on finding the most suitable matrices to encapsulate zerumbone and improve its low aqueous solubility, poor absorption, and low bioavailability. Therefore, in the present review, we gathered information on the natural sources of this compound, its antimicrobial activity, mainly associated with anti-biofilm action, and the latest strategies reported to improve the bioavailability of zerumbone and maintaining or increasing its activity, with a particular focus on advances in oral administration. 

## 2. Natural Sources of Zerumbone

The monocyclic sesquiterpenoid zerumbone has been found mainly in plants of the Zingiberaceae family, principally *Z. zerumbet*, which is considered an invasive species in certain regions, such as Taiwan [35,36,37]. Zerumbone is the major component (Table 1) detected in the essential oil [3,38,39,40,41], ethanol (87.4–242.73 mg/g), acetone (64.58%) and methanol extracts (2.767–33.018 mg/g) of *Z. zerumbet* rhizome [8,42,43], as well as the principal compound detected in many *Z. zerumbet* varieties such as *darcyi* (69.9%) and Jor Lab ZB-103 (32.79%) [9,44]. In contrast to rhizome oil, the essential oil extracted from the aerial parts (leaf, stem and inflorescence) of *Z. zerumbet* is dominated by nerolidol, *trans*-phytol, β-caryophyllene, linalool, pinenes, with a lesser content of zerumbone [38,40]. In general, both genotype and environment conditions influence the amount of zerumbone in *Z. zerumbet* rhizomes [44,45,46]. Thus, the essential oil obtained from fresh rhizomes of *Z. zerumbet* yielded more zerumbone than the oils from dry rhizomes of *Z. zerumbet* [6]. Optimization of extraction methods is a fundamental step in obtaining natural bioactive compounds [39,47]. In this sense, optimization with microwave-assisted extraction was 17.9, 28.1 and 30.2% higher than sonication, reflux and Soxhlet, respectively [48]. Zerumbone constitutes the predominant compound of the rhizome essential oil of other *Zingiber* spp. This genus is well-known for its abundant essential oils [49]. This sesquiterpenoid accounts for more than half of the sample of *Z. spectabile* Griff. (59.1%) [50]. It confers antibacterial activity against multidrug-resistant strains of *E. coli*, *S. aureus* and *Pseudomonas aeruginosa* (0.19–0.38 mg/mL) [50]. The essential oil from the leaves is rich in other sesquiterpenes, β-caryophyllene (21.3%) and β-elemene (12.5%), providing different odor and weaker antibacterial activity [50]. Other authors disagree on the chemical composition of *Z. spectabile* rhizome oil [51]. In particular, terpinen-4-ol (23.7%), labda-8(17),12-diene-15,16-dial (24.3%), α-terpineol (13.1%) and β-pinene (10.3%) were the major components found in rhizome oil of *Z. spectabile* from Malaysia [51]. The hexanolic extract of *Z. spectabile* rhizomes does not contain zerumbone, but 1,1′-ethylenebisdecali predominates instead (41.3%) [52].

Zerumbone is also the principal component of the essential oil extracted from *Z. amaricans* BL. (40.70–65.06%) and *Z. aromaticum* Val. (31.45%) [53,54] species. Both share other secondary components, such as α-humulene, β-selinene and caryophyllene oxide [54].

The sesquiterpenoid zerumbone has also been detected in the essential oil of the rhizome of *Z. montanum* (J. Koenig) Link ex A. Dietr., where it constitutes one of the main components with relevant pharmacological activities (anticancer, anti-inflammatory, antioxidant, antimicrobial, gastric and liver protection) [55]. These results differ from those of other authors who reported a different chemical composition of the essential oil *Z. montanum*. Particularly, in the volatile oil obtained from fresh rhizomes of *Z. montanum* in India, the monoterpenes 4-terpineol (38.0%) and sabinene (10.0%) were the most abundant compounds [56]. Similarly, Rajkumari and Sanatombi detected trace amounts (0.0107 mg/g) of zerumbone in the essential oil of in vitro cell cultures and microrhizomes of *Z. montanum* [57]. This cyclic sesquiterpene was the major compound in the ethanol extract isolated from *Z. montanum* from Bangladesh, followed by five kaempferol derivatives [58]. The essential oil of *Z. ottensii* Valenton rhizomes contained principally terpene compounds (Table 1), among which zerumbone was the main constituent (24.73%), followed by terpinen-4-ol (18.75%), sabinene (15.19%) and β-pinene (7.95%) [59]. Although the essential oils rich in zerumbone have countless applications and benefits, especially as anticancer agents, excessive and prolonged use can produce cytotoxic effects, including embryotoxic and teratogenic effects [59,60,61].

Zerumbone is not always one of the main components of *Zingiber* spp. rhizome essential oils [78]. For instance, zerumbone constitutes a minor component in the rhizome oils of *Z. cassumunar* Roxb. (0.4%) and *Z. officinale* Roscoe (1.25%) [79,80]. In other species, this molecule does not even appear. This is the case of essential oil obtained from the fresh rhizome of *Z. wrayi* var. *halabala* in Thailand, Thailandese *Z. kerrii* Craib and *Z. barbatum* Wall in Myanmar [81,82,83]. These differences between species may be due to the fact that the qualitative and quantitative chemical composition is subject to the origin, location, extraction methods, and degree of dryness of rhizomes [83,84]. As a consequence, the presence of zerumbone in *Z. zerumbet* rhizome oil constitutes a distinctive feature that allows the identification and differentiation from other ginger oils as well as different parts of the plant [38].

In the Zingiberaceae family, with approximately 53 genera and over 1200 species distributed in all tropical regions of the world, there are also rhizomatous genera such as *Curcuma*, well-known for their distinctive aroma and numerous bioactivities [85,86,87,88]. This shares certain bioactive compounds and medicinal properties with *Zingiber* spp [89]. Related to this, Xiang et al. studied the chemical composition and bioactivities of essential oils obtained from Chinese *Curcuma* spp., detecting zerumbone as one of the major components among them (1.08~15.45%) and the principal one in *C. rubescens* Roxb rhizome oil (15.45%) [71]. In contrast, Zhang et al. reported zerumbone as the second major compound (6.14 ± 1.42 mg/g) in *C. rubescens* essential oil, after ar-turmerone (6.88 ± 2.01 mg/g) [70]. Rhizome oil obtained from different Indian cultivars of *C. longa* L. also yielded high zerumbone content (0.383–8.333 mg/g), especially in dry conditions (3.209–8.333 mg/g), conferring promising immunomodulatory activity [90]. On the other hand, the essential oil of other *Curcuma* spp. contains lower quantities of zerumbone, such as *C. aromatica*, *C. caesia* and *C. xanthorrhiza* [91]. Other examples are *C. kwangsiensis* var *nanlingensis* essential oil that generated 0.32 mg/g of zerumbone in comparison to the main component β-elemenone (5.03 mg/g) [70], the volatile oil of *C. angustifolia* Roxb. yielded 2.10–2.17% of zerumbone with respect to the major compound epicurzerenone (29.60–31.18%) and rhizome oil of *C. zedoaria* (Christm.) Rosc. contained traces of the cyclic sesquiterpene (0.03%), while γ–eudesmol acetate was the main molecule (15.65%) [92]. Other abundant compounds in the essential oil of *C. zedoaria* were curcumin, ethyl *p*-methoxycinnamate and β-turmerone [93]. Despite being in lower concentration, zerumbone plays an important role in the bioactivity of these species. The minor components enhance the activity of the essential oil by means of synergism between them [72]. Zerumbone was also detected in the eluted extracts of the rhizome of *Curcuma* spp. It was one of the most significant compounds in the extracts of *C. caesia* Roxb. and *C. zedoaria*, showing potential anticancer and antioxidant activity, respectively [94,95]. 

*Alpinia* represents the largest, most extensive, and most taxonomically complex genus in Zingiberaceae family, with 230 species distributed throughout tropical and subtropical Asia. These plants live in forests and have been used as ornamentals, food and as a source of medicines due to their antimicrobial, cytotoxic or antioxidant activities [96,97,98]. The most cultivated and well-known species of this group is probably *A. galanga* (L.) Willd (AG) [96]. It represents a key ingredient in nutraceutical and pharmaceutical products due to the numerous benefits of its phytoconstituents, such as anti-inflammatory, anticancer and antioxidant effects [99]. Zerumbone is one of these functional components. It was the major compound detected in the essential oil of *A. galanga* from Sri Lanka (44.9%). In fact, it constitutes a new chemotype, different from other accessions in Malaysia, India and Indonesia that contain zerumbone in lower quantities [74,100,101]. The cyclic sesquiterpene was also detected in the methanolic extract in considerable amount (12.6 g) [102]. On the contrary, the essential oils resulting from other *Alpinia* spp. were richer in oxygenated monoterpenes and monoterpene hydrocarbons, such as 1,8-cineole, ocimenes, terpinen-4-ol and pinenes [103,104].

Zerumbone is also present in other genera included in Zingiberaceae family. The *Boesenbergia* genus is one of them [105]. In particular, the essential oil of *B. quangngaiensis* N.S. Ly, has good antibacterial and antifungal activity, and contains considerable concentration of zerumbone (11.4%) [75]. However, the chemical composition varies between species and parts of the plant. For example, zerumbone was absent in the leaf oil of other *Boesenbergia* spp., such as *B. armeniaca* and *B. stenophylla*, which were richer in other oxygenated sesquiterpenes (nerolidol, 42.55%), and oxygenated monoterpenes (linalool, 11.63%) and phenylpropanoids (methyl cinnamate, 83.17%), respectively [106]. Other species included in the genus *Boesenbergia* are also lacking in zerumbone, such as *B. pandurata* with mainly monoterpenes, mostly camphor (57.97%) [53]. The ethanolic extract of the Indonesian *Etlingera acanthoides* A.D. Poulsen also contains zerumbone [107], which accounts for 1.70% of the total essential oil. Together with the other main compounds, ar-turmerone, caryophyllene and caryophyllene oxide, zerumbone plays an important role in the antiviral activity of the plant [107]. 

Not only is zerumbone widespread in the Zingiberaceae family, but it represents one of the major elements of the essential oil of *Cheilocostus speciosus* (J. Konig) C. Specht [72,73]. Crape ginger, as it is popularly known, is a species belonging to the Costaceae family, closely related to Zingiberaceae. It is indigenous to peninsular Malaysia in southeast Asia from where it was introduced and naturalized in tropical and subtropical areas [108]. In general, phytochemicals extracted from different parts of this plant have shown numerous beneficial properties, including antioxidant, antidiabetic, antihypercholesterolemic, anticancer, hepatoprotective and anti-inflammatory effects [108]. In particular, the essential oil obtained from an Indian cultivar of *C. speciosus* showed anticancer activity, as well as considerable antibacterial capacity, against a wide range of Gram-positive (*S. aureus*, *B. subtilis*, *S. faecalis* and *S. albus*) and Gram-negative (*E. coli*, *P. aeruginosa*, *Klebsiella aerogenes*, *P. vulgaris*) bacteria. This feature is due to the synergism between the main components zerumbone (55.11%) and α-humulene (20.55%) [72]. Considering the data, it is thought that zerumbone is mainly localized in the rhizome of the plant species. Nevertheless, this monocyclic sesquiterpene has also been detected in different parts of other plants. Specifically, zerumbone was found to be one of the main components of the essential oil hydrodistilled from leaves of *Amomum gagnepainii* T.L.Wu, K.Larsen and Turland accounting for 16.4%, together with farnesyl acetate (18.5%) and β-caryophyllene (10.5%) [76]. 

## 3. Zerumbone Derivative

Zerumbone represents a point molecule for obtaining advantageous derivates. It is a sesquiterpenoid, specifically a cyclic (1E,4E,8E)-alpha-humulene ketone substituted with an oxo group at the carbon atom attached to two double bonds. This arrangement has been of central focus of research due to its unique reactivity in producing multiple structures as well as its numerous advantageous properties [36,109]. For years, researchers have examined and rearranged its structure with the aim of finding novel analogues with equal or greater activity than the original molecule [109,110,111,112]. This is the case with azazerumbone 2, a major Beckmann rearrangement product of *E*-zerumbone oxime with higher antibacterial and antimutagenic activity than zerumbone, zerumbone oxime, or zerumbol, possessing comparable or better bioactive attributes than the original sesquiterpene [112]. In addition, bio-isosteric replacements in the ortho position of the phenyl ring of zerumbone led to a series of new hydrazones, including 5a-f and 9a-f, with confirmed cytotoxic effects against three human cancer cell lines, HepG-2, SK-LU-1, and MCF-7 [111]. Another derivative, zerumbone epoxide, results from numerous reactions triggered by transition metal/Lewis acid-catalysed 1,4-conjugate addition of zerumbone [113]. 

In addition to the production of interesting derivatives, the study of zerumbone chemistry can help to expand the knowledge of sesquiterpenoids and their possible biosynthetic pathways [109]. Despite these findings, research on zerumbone production in suspension culture, as well as metabolic engineering, should be optimized to elucidate the biosynthetic pathway of this sesquiterpene and enhance the production of its derivatives [114]. 

## 4. Further Uses of Zerumbone 

The beneficial properties of zerumbone extend to other areas, such as the agri-food industry [40] (Figure 2). For example, Keerthi et al. observed that the concentration of zerumbone increases in *Z. zerumbet* when infected with the rot-causative necrotrophic phytopathogen *Pythium myriotylum*, exhibiting natural resistance against the contamination [115] (Figure 2). Furthermore, this molecule has shown promising antifungal activity against the other phytopathogenic fungi *Aspergillus* spp. (MIC 150–170 ppm), *Rhizoctonia solani* (EC_50_ 3906 ppm), *Sclerotium rolfsii* (EC_50_ 59.3 ppm) and *Macrophomina phaseolina* (EC_50_ 147.4 ppm) [40,64] (Figure 2). These plant-pathogenic fungi harm plants by secreting virulence factors such as effector proteins, degrading enzymes, toxins and growth regulators that severely affect crop yield and quality, and consequently have a negative impact on agriculture [116,117]. Amongst them, the essential oil of *Z. zerumbet*, rich in zerumbone (51.3%), has been especially effective against *Aspergillus* spp. [67]. The contamination of food and feed by *Aspergillus* has become a global issue with significant worldwide impact [118]. In particular, derived mycotoxins contaminate feed and food chains, being found in dairy products, dried fruits, vegetables and grain products, and representing a serious problem for public health [119,120]. 

In general, zerumbone and its analogues have demonstrated antibacterial and antimicrobial activities useful in food preservation [112]. *B. quangngaiensis* rhizome oil rich in zerumbone (11.4%) inhibited the growth of the spoilage bacteria *P. aeruginosa, E. coli and S. enterica* [75] (Figure 2). These microorganisms are able to contaminate meat, dairy and vegetable products, representing a considerable hazard for the agri-food industry and human health [121,122]. 

*Curcuma*-based botanicals have been evaluated for combating pests and pathogens of food crops [123]. Nowadays, they are emerging as sustainable alternatives to synthetic pesticides and natural protectants of crops [124,125,126]. In particular, the insecticidal activity of numerous essential oils has been confirmed [127,128]. Regarding our topic, the rhizome oil of *C. speciosus,* rich in zerumbone (38.6%), has exhibited interesting ingestion toxicity (LC_50_ 17.16 µg/mL) and ovicidal activity (EC_50_ 59.51 µg/mL) against the Old-World bollworm (*Helicoverpa armigera* (Lepidoptera:Noctuidae)), a polyphagous pest of increasing interest due to the rapid development of resistance to synthetic insecticides [73] (Figure 2). Furthermore, Wu et al. reported that the zerumbone analogue, α-caryophyllene (8.6%) exhibited higher insecticidal toxicity (LD_50_ 13.1 µg/adult) against *Lasioderma serricone*, a beetle usually affecting perishable food products such as cereals and legumes, than the own zerumbone (40.2%) (LD_50_ = 42.4 µg/adult) [10]. 

Zerumbone has also been reported to have a herbicidal effect (Figure 2). Weeds represent the most important biotic threat to agricultural production worldwide, accounting for the highest losses in crop productivity. This situation is expected to worsen as climate change progresses and resistance increases. Natural and bio-herbicides represent one strategy to combat weeds with less chemical harmful products [129,130,131]. Specifically, the monocyclic sesquiterpene showed an inhibitory effect on weed seed growth against *Phalaris minor* with no, or low, effect on germination, seed root and shoot length of *Triticum aestivum* [40,64] (Figure 2). 

The traditional uses of plant-based products have been studied over the years, and their numerous biological properties have been verified. In the field of agronomy, these compounds constitute economical, non-toxic and, consequently, environmentally friendly pesticides for sustainable agriculture. They represent a green alternative to synthetic pesticides, which are hazardous to living beings and the environment [132,133,134]. Among green chemicals, zerumbone, the main component of ginger plants, has demonstrated phytotoxic, antimicrobial, and insecticidal effects. Application of plant growth regulators (auxin and cytokine) with elicitors (methyl jasmonate and salicylic acid) to the fresh rhizome has been demonstrated to enhance the concentration of zerumbone and, consequently, its activity [135]. The food and feed preservative properties of zerumbone provide a basis for its use in the agri-food industry, to avoid contamination of harvests and stored functional food.

## 5. Formulation Strategies to Enhance the Solubility and Bioavailability of Zerumbone for Oral Administration

Even though the interesting properties of zerumbone have been widely demonstrated, zerumbone bioavailability and transport after oral administration is limited due mainly to its poor solubility. Zerumbone is freely soluble in organic solvents such as ethanol and dimethyl sulfoxide, while its solubility in water is about 1.296 mg/L at 25 °C. So, different approaches have been described during the years to improve the pharmacokinetics of low-water soluble drugs, such as particle size reduction, chemical modifications, amorphisation, salt formation, and through different strategies of formulation development such as solid dispersions, inclusion complexes, lipid-based formulations, micelles and nanoparticles [136,137]. These different formulation methods described in the literature to improve the solubility and bioavailability of zerumbone have recently been reviewed [138]. This section will, therefore, focus on updating the latest publications on the subject, with a particular focus on advances in oral administration. Additional information on the systems described in this section is summarized in Table 2. 

One of the strategies explored by several researchers to improve the saturation solubility and dissolution profile of water-insoluble compounds such as zerumbone, is the formation of inclusion complexes with cyclodextrin derivatives [22,139]. However, it is worth noting that while the effectiveness of hydroxypropyl-β-cyclodextrin (HP-β-CyD) in improving zerumbone solubilisation and cytotoxic activity in human cancer cell lines has been reported by several authors, there is limited information on the usefulness of other cyclodextrin (CyD) derivatives in improving the pharmaceutical properties of zerumbone [139]. Recently, Hassan et al. evaluated various CyD derivatives, such as zerumbone solubilising agents, along with their release profile and cytotoxic activity [33]. Among the different CyDs tested, sulfobutylether (SBE)-β-CyD showed the highest solubilizing effect for zerumbone, followed by the dimethyl (DM)-β-CyD and methyl (M)-β-CyD. It is interesting to note that they showed a remarkable solubilizing effect beyond complexation with (HP)-β-CyDs, and its supersaturation was maintained for several hours. Supersaturating drug delivery systems have been acknowledged as a promising concept to obtain adequate oral bioavailability [143]. Regarding their cytotoxic activity, while an SBE-β-CyD/Zerumbone complex retained the activity of Zerumbone, M-β-CyD and DM-β-CyD showed toxicity compared to HP-β-CyD. Finally, to better understand the release mechanism of zerumbone from the different CyDs, crystallographic studies and molecular simulations were used to demonstrate the stronger interactions of SBE-β-CyD with zerumbone compared to HP-β-CyD (binding energy of −18.72 kcal/mol for SBE-β-CyD/Zerumbone vs. −16.24 kcal/mol for HP-β-CyD/Zerumbone).

Despite its promising pharmacological activities, preclinical and clinical studies of zerumbone have been limited due to its poor aqueous solubility, poor absorption, and low oral bioavailability. One example of these few studies involved the use of an inclusion complex of zerumbone with hydroxypropyl-β-cyclodextrin to treat acute and chronic gastritis. All patients of the vehicle group were injected with 20 µM doses of zerumbone over a 20-day period [22]. After evaluation through endoscopic studies, histopathological examinations, and urea breath tests, it was concluded that zerumbone treatment was quite effective in acute and chronic gastritis with or without *H. pylori* infection. This study once again confirms the strong potential of zerumbone as a therapeutic agent for various diseases, even including acute and chronic settings [4,144,145]. However, to our knowledge, no preclinical or clinical trials have been reported on the efficacy of formulations developed for the oral administration of zerumbone. Further work in this area, including investigations of the corresponding pharmacokinetic and pharmacodynamic data in humans, is therefore required.

Loading zerumbone into lipid carriers is another commonly reported approach to improve its solubility and delivery by facilitating transport across several anatomical barriers, thus avoiding mucosal toxicity. Specifically, nanostructure lipid carriers (NLC) have demonstrated highly advantageous characteristics, including low toxicity, controlled release, biodegradability, high drug loading, low cost, and easy upscaling [31,146,147]. Other proposed advantages of using NLC encapsulation as an oral delivery system are that they are transported via the lymphatic system avoiding first-pass metabolism, thus achieving a higher peak concentration after oral administration. In addition, NLC systems have shown greater permeability in the duodenum, jejunum, ileum and colon, improving absorption in the gastrointestinal tract post oral administration [148]. 

NLC–Zerumbone has been reported to possess in vitro cytotoxicity against several types of cancer cell, and no observed in vivo oral acute toxicity [31,140,149,150]. For example, Hosseinpour et al. evaluated the anticancer effect of zerumbone and zerumbone-loaded NLC on the human mammary gland adenocarcinoma (MDA-MB-231) cell line [147]. The efficacy of not-loaded and NLC-loaded zerumbone in suppressing the proliferation of MDA-MB-231 was confirmed by electron and fluorescent microscopy and flow cytometry (IC_50_ of 5.96 ± 0.13 and 6.01 ± 0.11 μg/mL for zerumbone and zerumbone-NLC, respectively). Furthermore, both systems significantly induced apoptosis via activation of caspase-9 and caspase-3 inhibition of antiapoptotic protein, and stimulation of proapoptotic protein expressions in a time-dependent manner. Further information on the in vivo effects of orally administrated NLC-zerumbone was reported in a study by Akhtar et al. in which the antitumor, immunostimulatory and anti-inflammatory effects of zerumbone and NLC-zerumbone in in vivo 4T1 challenged mice were compared [8]. NLC-zerumbone was found to further improve the efficacy of zerumbone, effectively controlling tumor growth and metastasis via delaying cancer cell cycle progression and apoptosis. In addition, zerumbone encapsulation was able to better contain the progression of 4T1 breast cancer by controlling cancer cell mitosis, inducing apoptosis, activating anti-tumor immunity, controlling inflammation in the tumor microenvironment, and delaying tumor metastasis. 

Zeolite Y-gelatin composites have also been reported to be interesting oral delivery carriers of zerumbone, providing effective sustained release [142]. The ability to release the compound over an extended period of time at a controlled rate is generally more effective and safer, since it overcomes the possible toxicity associated with the drastic accumulation of the drug in the body over a short period. The design and preparation of cross-linked zeolite Y/gelatine composite was developed using glutaraldehyde as cross-linking agent targeting to prolong the duration of zerumbone released for 24 h [141]. The solid state of zerumbone after loading remained crystalline, ensuring no form changes during storage. Furthermore, cross-linking improved the thermal stability and the composite showed resistance to degradation under acidic and basic conditions (pH 1.2 and pH 7.4), confirming the potential of this system for oral administration. Despite the promising in vitro zerumbone controlled release, further in vivo antitumor evaluation of these matrices is necessary.

## 6. Conclusions

The agri-food industry requires natural pesticides, including essential oils and/or their main compounds, as substitutes for synthetic pesticides that have adverse effects on soils, water, wildlife, human health, and the environment. Natural compounds (monoterpenoids and sesquiterpenoids) present in essential oils could be used as preservatives in the food industry, with additional benefits due to their antioxidant properties, as well as their taste or aroma characteristics. Among the bioactive compounds, sesquiterpenoids have remarkable value due to their prominent presence in spices and herbs, and their wide range of activities due to their structural variability. In this sense, *Z. zerumbet*, whose main compound is zerumbone, is a commercialized spices widely consumed by humans. However, the bioavailability and biodegradation of these products are important properties influencing the half-life time and possible persistence problems in the human body. In this sense, zerumbone’s encapsulation represents an interesting alternative to solve the problems of solubility, oxidation, and other types of deterioration due to environmental factors such as light, temperature or humidity. Nonetheless, to date, practical applications of encapsulation and zerumbone-controlled delivery systems are scarce. Considering the relevance of this multifunctional sesquiterpene, further preclinical and clinical studies are needed to demonstrate and evaluate its in vivo efficacy, as well as formulation studies to fully understand its potential application as a therapeutic agent, green pesticide and/or food preservative.

## Figures and Tables

**Figure 1 plants-12-00005-f001:**
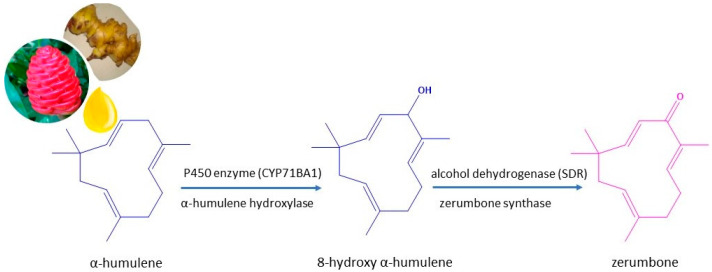
Biosynthetic route of zerumbone from α-humulene catalyzed by a short-chain dehydrogenase/reductase (SDR).

**Figure 2 plants-12-00005-f002:**
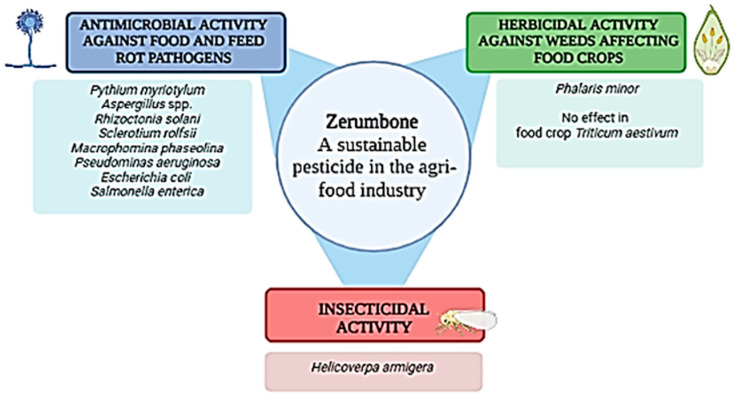
Applications of zerumbone in the agri-food industry.

**Table 1 plants-12-00005-t001:** Natural sources of zerumbone according to the plant species, origin, method of extraction and analysis. RP-HPLC-PDA: reverse phase high performance liquid chromatography, GC-MS: gas chromatography-mass spectrometry, FID: flame-ionization detector, HPLC: high performance liquid chromatography, FTIR: Fourier transform infrared spectroscopy, UHPLC: ultra-high performance liquid chromatography, HRESI: high-resolution electrospray ionization mass, EI-MS: electron ionized-mass spectroscopy, ^1^H-NMR: (nuclear magnetic resonance) and ^13^C-NMR.

Plant Species	Plant Part	Origin	Method of Extraction	Analysis	Yield	Main Components	Activity	Ref.
*Zingiber zerumbet*	Rhizome	India	Methanol extraction	RP-HPLC-PDA		Zerumbone (33.02 mg/g)	Antioxidant	[42]
	Rhizomes	India		GC/FID- GC/MS		Zerumbone (70.60%) α-Humulene (5.65%) Humulene epoxide I (5.21%) Humulene epoxide II (5.71%) Camphor (1.90%) Camphene (3.47%) Caryophyllene oxide (2.52%)	Antioxidant	[41]
	Rhizomes	India	Hydrodistillation Clevenger	GC- GC/MS	0.45%	Zerumbone (74.82%) Humulene (6.02%) β-Copaen-4α-ol (4.32%)	Antifungal	[40]
	Rhizomes	Brazil	Hydrodistillation Clevenger	GC/MS	5%	Zerumbone (87.93%)	Antibacterial	[3]
	Rhizomes	Malaysia	Hydrodistillation		0.25%		Anti-inflammatory	[62]
	Rhizomes	Indonesia	Water distillation	GC/MS	0.12%	Sabinene (32.96%) β-Myrcene (13.27%) Zerumbone (11.05%)	Increase body weight	[63]
	Rhizomes	Malaysia	Hydrodistillation Turbo Extractor Distillator	HPLC	0.35%	Zerumbone (126.54 mg/mL)	Antibacterial	[39]
	Rhizomes	Reunion Island	Steam distillation	GC-GC/MS- GC/FTIR	0.3–0.4%	Zerumbone (36%) α-Humulene (14.4%) Camphene (13.8%) Caryophyllene oxide (5.2%) Camphor (3.8%) 1,8-Cineole (3.2%)		[38]
	Rhizomes	Malaysia	Ethanol extraction	HPLC		Zerumbone (242.73 mg/g)	Immunosuppressant	[8]
	Rhizomes	Malaysia	Microwave	UHPLC		Zerumbone (4.82 mg/g DM)	Antiproliferative	[48]
	Rhizomes	India	Methanol extraction	RP-HPLC		Zerumbone (15.598–30.143 mg/g)	Antioxidant	[42]
	Fresh rhizomes	China	Hydrodistillation Clevenger	GC/MS– GC/FID		Zerumbone (40.2%) α-Caryophyllene (8.6%) Humulene epoxide II (7.3%) Camphene (5.9%)	Insecticidal, repellent	[10]
	Fresh rhizomes	China	Hydrodistillation Clevenger	GC-FID/MS	0.65%	Zerumbone (75.0%) α-Humulene (6.5%) Humulene oxide I (3.8%) Camphene (3.3%) Humulene oxide II (2.7%) Camphor (1.3%) Caryophyllene oxide (1.3%) 1,8-Cineole (1.2%)	Antibacterial, cytotoxic	[6]
	Fresh rhizomes	India	Hydrodistillation Clevenger	GC/MS	1.12%	Zerumbone (49.3%) α-Caryophyllene (20.1%) Z-Caryophyllene (3.8%)	Antifungal, antimycotoxin	[64]
	Fresh rhizomes	Malaysia	Hydrodistillation	GC-FID/MS	0.25%	Zerumbone (36.12%) Humulene (10.03%) Humulene oxide I (4.08%) Humulene oxide II (2.14%) Caryophyllene oxide II (1.66%) Caryophyllene oxide I (1.43%)	Antinociceptive	[65]
	Fresh rhizomes	Vietnam	Steam distillation	HPLC	0.1%	Zerumbone (98%)	Weak in vitro vasodilator	[66]
	Fresh rhizomes	India		GC/MS	0.75%	Zerumbone (32.79%) Camphene (19.41%) Eucalyptol (6.80%)		[44]
	Dried rhizomes	China	Hydrodistillation Clevenger	GC-FID/MS	0.39%	Zerumbone (41.9%) α-Humulene (29.4%) Humulene oxide I (6.0%) Humulene oxide II (3.9%) Camphene (3.9%) β-Caryophyllene (2.5%) Camphor (2.4%) Caryophyllene oxide (2.1%) 1,8-Cineole (1.2%)	Antimicrobial, cytotoxic	[6]
	Dried rhizomes	Malaysia	Hydrodistillation Clevenger	GC-GC/MS	0.37%	Zerumbone (58.44%) α-Humulene (12.24%) Camphene (5.36%)	Immunosuppressant	[8]
	Air-dry rhizomes	Vietnam	Hydrodistillation Clevenger	GC/MS	0.65%	Zerumbone (51.3%) Humulene epoxide I (6.4%) Humulene epoxide II (5.5%) α-Humulene (5.4%) Camphene (4.1%) 1,8-Cineole (3.2%)	Insecticidal, antifungal	[67]
	Powdered rhizomes	India	Acetone extraction	HPLC		Zerumbone (99.94%)	Anticancer	[68]
	Dried powdered rhizomes	India	Acetone extraction Soxhlet	GC/MS	2.86%	Zerumbone (64.58%) Diacetone alcohol (10.64%) α-Humulene (8.93%) Caryophyllene oxide (5.68%) Humulene epoxide (3.18%)		[43]
	Powdered rhizomes	Malaysia	Ethanol extraction	HRESI/MS		Zerumbone (87.4 mg)	Immunosuppressant	[69]
	Roots	India	Methanol extraction	RP-HPLC-PDA		Zerumbone (05.562 mg/g)	Antioxidant	[42]
*Z. zerumbet* var. *darcyi*	Fresh rhizomes	India	Hydrodistillation Clevenger	GC-GC/MS	0.23%	Zerumbone (69.9%) α-Humulene (12.9%) Humulene epoxide II (2.5%) Caryophyllene oxide (1.1%) Camphene (1.9%)		[9]
*Z. ottensii*	Fresh rhizome	Thailand	Hydrodistillation Clevenger	GC/MS	0.24%	Zerumbone (24.73%) Terpinen-4-ol (18.75%) Sabinene (15.19%) β-Pinene (7.95%)	Cytotoxic	[59]
	Fresh rhizomes	Thailand	Hydrodistillation	GC/MS	0.21%	Zerumbone (25.21%) Sabinene (23.35%) Terpene-4-ol (15.97%)	Apoptotic	[61]
*Z. montanum*	Dried rhizomes	Bangladesh	Ethanol extraction	^1^H-NMR/^13^C-NMR		ZerumboneFlavonoid derivatives		[58]
*Z. amaricans*		Malaysia	Hydrodistillation	GC-MS		Zerumbone (40.70%)		[53]
	Powdered dried rhizomes	Indonesia	Steam distillation	GC-MS	1.6%	Zerumbone (65.06%) Humulene oxide (9.66%) α-Humulene (9.41%)		[54]
*Z. aromaticum*	Powdered dried rhizomes	Indonesia	Steam distillation	GC-MS	2.8%	Zerumbone (31.45%) Cyclohexene (13.52%) Isogeraniol (10.52%)		[54]
*Curcuma rubescens*	Fresh rhizomes	China	Steam distillation	GC/MS	4.36%	Zerumbone (6.88%) Germacrone (4.99%)		[70]
	Powdered dried rhizomes	China	Hydrodistillation Clevenger	GC/MS		Zerumbone (15.45%) *ar*-Turmerone (13.80%)	Antimicrobial, antioxidant, anticancer, anti-inflammatory	[71]
*Cheilocostus speciosus*	Rhizomes	India	Steam distillation	GC-MS/FID	0.18 g	Zerumbone (55.11%) α-Humulene (20.55%)	Anticancer, antibacterial	[72]
	Fresh rhizomes	India	Hydrodistillation Clevenger	GC/MS	1.9 mL/kg	Zerumbone (38.6%) α-Humulene (14.5%) Camphene (9.3%)	Insecticidal	[73]
*Alpinia galanga*	Dried crushed rhizomes	Sri Lanka	Hydrodistillation Clevenger	GC/MS	0.56%	Zerumbone (44.9%)		[74]
*Boesenbergia quangngaiensis*	Rhizomes	Vietnam	Hydrodistillation	GC/MS	0.16%	*cis*-β-Elemene (18.4%) Zerumbone (11.40%) Myrtenyl acetate (10.6%)	Antimicrobial	[75]
*Amomum gagnepainii*	Leaves	Vietnam	Hydrodistillation	GC-MS/FID	0.20%	Farnesyl acetate (18.5%) Zerumbone (16.4%) β-Caryophyllene (10.5%)		[76]
*Cyperus rotundus*	Air-dried rhizomes	South Korea	Methanolextraction	EI-MS/^1^H-NMR/^13^C-NMR		Zerumbone 1,8-Cineole	Insecticidal, repellent	[77]

**Table 2 plants-12-00005-t002:** Additional information on the studies reviewed in this section.

Technique	Polymer/Matrix	Solubility Enhancement	Key Results	Reference
Inclusion complexes	HP-β-CyD	>30 fold (From 0.0053 mM for free zerumbone to 0.173 mM with 0.01 M HPβCyD at 20 °C)	Molecular modelling calculations showed that zerumbone penetrates completely into the cavity of the HPβCyD. Complexation efficiency zerumbone-HPβCyD was of 1.04. In vitro cell survival assay on human cervical cancer (Hela), breast cancer (MCF7 and MDA-MB 231) and human leukemic (CEMss) cell lines.	[139]
Inclusion complexes	HP-β-CyD	-	Clinical study conducted on three groups of 220 patients each) to treat acute and chronic gastritis. Patients injected with 20 µM of zerumbone.	[34]
Inclusion complexes	HP-β-CyD	8 mM of zerumbone dissolved in 50 mM of HP-β-CyD 18 mM zerumbone dissolved in 18 mM of SBE-β-CyD	CyD derivatives are useful to maintain the supersaturation state of zerumbone	
	SBE-β-CyD			[33]
Nanostructured lipid carrier	Palm oil:Olive oil:Lipoid S100 (7:3:3)	Entrapment efficiency of 99.03% Drug loading of 7.92%.	Half maximal inhibitory concentration (IC_50_) of ZER-NLC was 5.64 ± 0.38 μg/mL (free zerumbone IC50 was 5.39 ± 0.43 μg/mL) after 72 h of treatment. Zerumbone released after 48 h from NLC was 46.7% (0.95 mg) vs. 90.59% (1.81 mg) from pure drug dispersion. In vitro cytotoxicity tests in Jurkat T-cell line.	[31]
Nanostructured lipid carriers	Palm oil:Olive oil:Lipoid S100 (7:3:3)	Entrapment efficiency of 99%	Acute toxicity study for zerumbone-NLC conducted orally in BALB/c mice (single dose for 14 days. Oral doses of 100 and 200 mg/kg showed no sign of toxicity or mortality. LD50 is higher than 200 mg/kg therefore safe by oral administration.	[140]
Nanostructured lipid carriers	Palm oil:Olive oil:Lipoid S100 (7:3:3)	Entrapment efficiency of 99%	Release of environ 50% after 48 h. In vivo tests conducted on mice challenged with 4T1 breast cancer.	[29]
Pore Encapsulation	Zeolite Y-gelatin	Entrapment efficiency tests were here performed by soaking 2% *w/v* zeolite Y in 5, 10 and 15% aqueous solution of 100 µM zerumbone. EE% of 93.7, 97.3 and 99.2% for the zerumbone concentration of 5, 10 and 15% respectively.	In vitro controlled release tests confirm the interest of this system for oral administration	[141,142]

## Data Availability

Not applicable.

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
