# Peer review of "Healthy Zerumbone: From Natural Sources to Strategies to Improve Its Bioavailability and Oral Administration"

_plants, 2022, doi:10.3390/plants12010005_

Round 1

Reviewer 1 Report

The review article entitled: Healthy Zerumbone: from natural sources to strategies to improve its bioavailability and oral administration, is interesting, well written and will contribute to the field of research in the area of developing new drugs and pesticides based on natural products.

Author Response

The review article entitled: Healthy Zerumbone: from natural sources to strategies to improve its bioavailability and oral administration, is interesting, well written and will contribute to the field of research in the area of developing new drugs and pesticides based on natural products.

Thank you very much for reviewing our manuscript.

English language and style (x) English language and style are fine/minor spell check required

Thank you very much for your observation. The misspellings have been corrected.

Reviewer 2 Report

Zerumbone is a monocyclic sesquiterpenoid  with potential  antimicrobial, antitumor, hyperalgesic,  antioxidant and anti-inflammatory applications, isolated from wild ginger (Zingiber zerumbet). The article is interesting and can be published in Plants after a few minor revision>

Zerumbone chemical structure and physico-chemical properties should be presented, The article should emphasize that the therapeuticall application of the substance are potential, as no pharmaceutical preparation was approved by authorities. 

Author Response

Zerumbone is a monocyclic sesquiterpenoid with potential antimicrobial, antitumor, hyperalgesic, antioxidant and anti-inflammatory applications, isolated from wild ginger (Zingiber zerumbet). The article is interesting and can be published in Plants after a few minor revision. Zerumbone chemical structure and physico-chemical properties should be presented, The article should emphasize that the therapeuticall application of the substance are potential, as no pharmaceutical preparation was approved by authorities. 

Thanks for analyzing our manuscript and for your comments. We have corrected our manuscript following your comments.

The chemical structure of zerumbone appears in Figure 1 and the physico-chemical properties have been added in different sentences throughout the manuscript:

Line 26: (C15H22O)

Lines 59-61: sesquiterpene of 65.3 ºC of melting point, contains in an 11-membered ring, an α, β-unsaturated carbonyl group, and three double bonds (two conjugated and one isolated)

Lines 369-370: Zerumbone is freely soluble in organic solvents such as ethanol and dimethyl sulfoxide, while its solubility in water is about 1.296 mg/L at 25 ºC. So,

The authors agree with the reviewer, but we had already indicated in the conclusion section (line 480) that the therapeutic applications of zerumbone were potential.

English language and style (x) Moderate English changes required

Thank you very much for your observation. The misspellings and some sentence have been corrected.

Reviewer 3 Report

Dear colleagues, I consider that the present study is an interesting review, with current information about the natural sources of zerumbone, its biotechnological applications and the types of formulated products. I believe that this review is a very good starting point for researchers interested in the development of technologies for the production of this compound, but also in the creation of new products based on zerumbone. Personally, I have only one specific comment regarding figure 1, L66, I think it should be more elaborated and with more explanations directly in the figure.

Author Response

Dear colleagues, I consider that the present study is an interesting review, with current information about the natural sources of zerumbone, its biotechnological applications and the types of formulated products. I believe that this review is a very good starting point for researchers interested in the development of technologies for the production of this compound, but also in the creation of new products based on zerumbone. Personally, I have only one specific comment regarding figure 1, L66, I think it should be more elaborated and with more explanations directly in the figure.

The authors appreciate the comment to improve the manuscript and we have modified figure 1, adding more information.

Figure 1. Biosynthetic route of zerumbone from α-humulene catalyzed by a short-chain dehydrogenase/reductase (SDR).

English language and style (x) English language and style are fine/minor spell check required

Thank you very much for your observation. The misspellings have been corrected.